# EV Charging in Case of Limited Power Resource

**Manan'Iarivo Louis Rasolonjanahary [1,\*], Chris Bingham [1], Nigel Schofield [2] and Masoud Bazargan [3]**

[1] School of Engineering, University of Lincoln, Brayford Pool, Lincoln LN6 7TS, UK; cbingham@lincoln.ac.uk
[2] School of Computing and Engineering, University of Huddersfield, Huddersfield HD1 3DH, UK; N.Schofield@hud.ac.uk
[3] Power Technologies Limited, Upper Reule Cottage, Stafford ST18 9JH, UK; masoud.bazargan@powertechnologieslimited.com
\* Correspondence: 17687516@students.lincoln.ac.uk

**Abstract:** In the case of the widespread adoption of electric vehicles (EV), it is well known that their use and charging could affect the network distribution system, with possible repercussions including line overload and transformer saturation. In consequence, during periods of peak energy demand, the number of EVs that can be simultaneously charged, or their individual power consumption, should be controlled, particularly if the production of energy relies solely on renewable sources. This requires the adoption of adaptive and/or intelligent charging strategies. This paper focuses on public charging stations and proposes methods of attribution of charging priority based on the level of charge required and premiums. The proposed solution is based on model predictive control (MPC), which maintains total current/power within limits (which can change with time) and imparts real-time priority charge scheduling of multiple charging bays. The priority is defined in the diagonal entry of the quadratic form matrix of the cost function. In all simulations, the order of EV charging operation matched the attributed priorities for the cases of ten cars within the available power. If two or more EVs possess similar or equal diagonal entry values, then the car with the smallest battery capacitance starts to charge its battery first. The method is also shown to readily allow participation in Demand Side Response (DSR) schemes by reducing the current temporarily during the charging operation.

**Keywords:** model predictive control; non-scheduled; power limited sources; electric vehicle; battery; scheduled and stop-start battery charging

## 1. Introduction

According to Hardman et al. [1], the widespread use of electric vehicles (EV) presents many benefits to the environment, but it can also pose significant operational challenges to existing power networks. This large-scale adoption of EVs could lead to uncontrolled charging and cause a range of power network problems, including shortage of power, voltage limit violations, component overloads, power system losses, phase imbalance, and issues with power quality and stability. These issues affect mainly distribution grids [2–4].

For these issues, mitigation requires the use of an appropriate EV charging control strategy. Most of the charging control strategies published in previous research can be classified into two categories. The first favors EV users and includes a strategy based on minimization of total charging cost or total peak consumption [5,6], a strategy aimed at minimizing the average waiting time and charging cost. The second, which could alleviate EV impacts on the grid, support its safe operation and includes a strategy encouraging temporary EV load shifting to reduce the overlap with the residential peak load periods [7] and a strategy designed to maximize EV integration. As indicated by its name, this strategy determines the maximum share of EVs that can be safely connected to the grid [8]. Strategies consisting in leveraging or nudging driver behavior using charging tariffs to prevent grid congestion and the minimization of energy losses in the distribution system

are also proposed, respectively, in [9,10]. Some control strategies benefit both EV users and the grid. This is the case of valley-filling charging proposed in [11], where EVs are charged during valley times according to the load profile to minimize energy losses. This is also the case of an intelligent charging method for EV charging facilities based on Time-of-use (TOU) pricing. The purpose is to alleviate the stress in the power grid under peak demand and to meet the demand response requirements in the regulated market [12].

For a shortage of energy, the solution can be achieved through the installation of additional support, such as battery storage, but this is not always practical or cost-effective. In [13], the use of biogas/biomass resources for charging EVs was investigated and found to be promising in the support of the electricity grid. In [14], Aziz et al. present a battery-assisted charging system that would minimize stresses and maintain the quality of grid electricity. This system consists of a large stationary battery. Despite the assistance of these storages, the shortage of energy could still be possible in the case of pure reliance on renewable energy. According to [15], the use of 100% renewable transportation is feasible, but not necessarily compatible with an indefinite increase in resource consumption. This can arise in a public station with multiple EVs and limited power sources during peak demand periods. A charging control strategy needs to be applied to mitigate the impacts on the system. For the case of resource constraints, the scheduling of the charging process of the EVs is introduced and discussed in this paper. Each EV was accorded a priority according to different criteria. This issue belongs to the field of "resource-constrained project scheduling problems (RCPSP)" in industrial and management projects [16]. Different solutions are proposed. These include the heuristic method [17] and linear programming. In [18], Jhala et al. apply Linear Programming in the coordinated charging of EVs from renewable energy sources in commercial parking, allowing for system constraints due to transformer limitations. In [19], Shamsdin et al. use linear programming in four policies, including random charging, lowest state-of-charge, and shortest parking time. These algorithms place some limitations on large problems and the quality of their solutions. In [20], Ren and Wang suggested the use of multi-agent methods to tackle project scheduling problems. Zheng and Wang present a method based on multi-agent systems and swarm intelligence to deal with RCPSP [21]. Ohtani et al. propose an algorithm based on multi-agent to deal with switching schedules in a system with the constraint on peak power [22]. In their method, the agents were provided with prescheduled switching patterns to deal with power constraints. Considering the fast response required by the grid for its stability, most of these methods can hardly be applied to control charging of EVs, for which the input (amount of charge required) is not predictable in advance in real-time. In this paper, a new control strategy to deal with supplies with an inherently limited power source (which can change in real-time) is proposed. An algorithm based on MPC associated with multi-agent is used. MPC is an important advanced control technique for applications where unaccommodated hard- and soft- constraints could readily make more traditional multivariable feedback systems impart closed-loop instability [23]. It has been successfully applied to many industrial control systems [24]. A further consideration is that during high-demand periods, the use of charging bays should not unduly affect local domestic/residential supplies. This paper, thus, focuses on example public EV stations with charge scheduling based on allocation priorities to each vehicle whilst also maintaining maximum total current/power constraints and being receptive to participation in DSR events when requested.

## 2. Problem Definition

The quantity of electricity fed into the grid should be equal to the amount of electricity consumed to prevent fault scenarios and blackouts [25]. With the increase in renewable energy production, balancing the grid is becoming increasingly complicated, with natural factors impacting stability. If demand exceeds supply, the grid frequency drops (and vice versa) and, without intervention, this can put sensitive equipment at risk and create a shutdown of generating units, further exacerbating the problem. It is important, therefore, that the additional consumption expected from the widespread charging of EVs, particu-

larly in islanded charging stations, is controlled so as not to overload other domestic and industrial supply lines. This paper therefore proposes a method using a multi-agent system associated with MPC to schedule EV charging operation subject to resource availability.

## 3. Model and Method

### 3.1. Battery Model

For modeling purposes, each EV battery is simply represented by a capacitance $C$ connected in parallel with a large resistance $r$ representing a small leakage, as in Figure 1. The capacitance value is derived from the equivalent energy capacity of a respective battery using (1), assuming that one starts from 79% of full capacity, where $E$ is its rated energy capacity measured in Joules, and $V_{ref}$ is the nominal fully charged nominal terminal voltage (leakage $r = 100$ k$\Omega$ in this case).

$$C = \frac{E}{0.188V_{ref}^2} \tag{1}$$

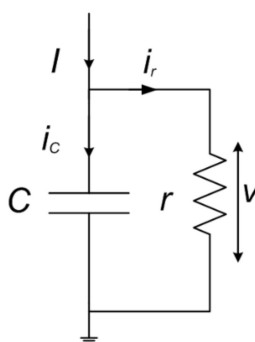

**Figure 1.** EV battery analogic model.

### 3.2. Problem Formulation

Charging multiple EV batteries under conditions of limited total power resources requires the solution of an optimization problem such that, for each car, the target battery's terminal voltage will be achieved over a particular time period. Here, this is formulated using the minimization of a cost function $J$ specified in the quadratic form (2):

$$J(v, I) = \frac{1}{2} \sum_{k=1}^{M} ||v(k) - v_{ref}(k)||_Q^2 + ||I(k) - I(k-1)||_R^2 \tag{2}$$

where

- $n$ is the number of EVs being charged,
- $v(v(1), v(2), v(3), \dots, v(n))$ and $I(I(1), I(2), I(3), \dots, I(n))$ are, respectively, the battery voltage and input charging current of the EVs battery,
- $M$ is the control horizon of the MPC, and
- $Q$ and $R$ are symmetric and positive definite matrices denoting, respectively, the output priorities and relative 'weights' that consider the impact of the input $I$. For the matrix $Q$, the larger the magnitude of the diagonal entry, the greater the corresponding priority.

The minimization of $J(v, I)$ is subject to the following constraints: Equation (3) which is the state-space model of the battery, and Equation (4) which specifies the constraint of total available power (or maximum charging current available to charge all vehicles at any time).

$$\frac{dv}{dt} = Av + BI \tag{3}$$

where $A$ and $B$ are, respectively, the state and input matrices of appropriate dimensions shown below:

$$A = \begin{pmatrix} -\frac{1}{C_1 r_1} & \cdots & 0 \\ \vdots & \ddots & \vdots \\ 0 & \cdots & -\frac{1}{C_k r_k} \end{pmatrix} \text{ and } B = \begin{pmatrix} \frac{1}{C_1} & \cdots & 0 \\ \vdots & \ddots & \vdots \\ 0 & \cdots & \frac{1}{C_k} \end{pmatrix}$$

and

$$\sum_{n=1}^{k} I_k \leq I_{lim} \tag{4}$$

Throughout, it is assumed that the terminal voltage of each battery is known/measured i.e., $v_k(t = 0) = vk0$ at the start of the charging operation.

The optimization problem can be readily solved dynamically when the continuous model (3) is transformed to a discretized form (5)

$$v[j+1] \approx (I_D + TA)\, v[j] + TBI_D[j] \tag{5}$$

where $T$ is the sampling period, $ID$ represents the identity matrix with the same dimension as $A$ and $j = 0, 1, 2, \ldots$ .

### 3.3. Model Predictive Control (MPC)

MPC is a well-established technique for advanced process control in many industrial applications, with the 'cost' penalized based on a quadratic function, Equation (2). It relies on the solution of a constrained optimization problem using a receding horizon approach at each time step and readily caters for multiple input, multiple output control problems with both hard and soft constraints. Here, the receding horizon control problem follows that of [26]:

- At time $k$ and for the current state $v(k)$, solve, on-line, an open-loop optimal control problem over some future interval of length $M$, taking into account the current and future constraints. $M$ is the length of the receding horizon.
- Apply the first step in the optimal control sequence to the plant, and
- Repeat the procedure at the time $(k+1)$ using the current state $v(k+1)$.

The main components of the MPC are:

- The plant model,
- An objective function,
- A state estimator, and
- An algorithm for solving constrained optimization problems.

At each iteration, the MPC performs the following process:

○ Measure the system outputs and inputs.
○ Estimate the current state.
○ Calculate the next control move by solving the optimization problem and applying the required control action.

### 3.4. Proposed Priority Attribution Strategies for EV Charging

Currently, most charging stations operate on a first-come/first-served basis. However, as the number of vehicles in use increases, this basic principle could lead to long queues, with limited numbers of bays. Ideally, therefore, public charging stations (e.g., at the shopping centres) should not be used for long-duration charging. Moreover, even if there are sufficient free bays at a public station, the simultaneous charging of all EVs may introduce some power limiting issues at peak times. To address this, here we proposed to attribute a priority to each charging EV by appropriate selection of the $Q$ matrix in an MPC cost function. The selection of cost function parameters ($Q$) can be chosen based on various desirable attributes, some of which are considered below.

### 3.4.1. Priority Based on the Level of the Charge Request

In this strategy, the matrix priority $Q$ is derived according to the inverse of the square of the charge $L$ required. This is to allow priority charging of EVs requiring the smallest amount of charge (i.e., top-up). Each car features its consumption performance, referred to as the ratio of distance travelled and energy consumption—examples for some common EVs are shown in Table 1. An estimate of the level of required charge $L$ is derived according to the required travel distance. For a public charging station, an example choice of the matrix $Q$ could therefore be:

$$[Q] \sim [L]^{-2} \tag{6}$$

The power $-2$ of the matrix $[L]$ was chosen from a trial simulation. It offers better conditioning of the quadratic form matrix $[Q]$.

**Table 1.** Car data [1].

| Car | Battery Capacity (kWh) | Distance Range (km) |
|---|---|---|
| Tesla | 100 | 515 |
| Jaguar I-Pace | 90 | 362 |
| Nissan Leaf | 62 | 217 |
| BMW i3 | 42.2 | 233 |
| Renault Zoe | 52 | 314 |

[1] https://ev-database.uk/car/ (accessed on 10 July 2021).

### 3.4.2. Priority Based on Premium

In this case, the matrix priority $Q$ is attributed according to the price $[P]$ that the customer is willing to pay. At a charging station, the price displayed is fixed, but it is suggested to give priority to customers who are willing to pay more than the nominal. Examples of prices at UK charging stations are shown in Table 2. According to this option, the matrix $Q$ could be chosen as:

$$[Q] \sim [P] \tag{7}$$

**Table 2.** Example of UK public pricing [1].

| Company Name | Standard (p/kWh) |
|---|---|
| Ecotricity | 30 |
| GeniePoint | 30 |
| Instavolt | 35 |
| Shell Recharge | 39 |
| Tesla | 25 |
| Ubitricity | 24 |

[1] https://www.zap-map.com/charge-points/public-charging-point-networks/ (accessed on 3 December 2020).

### 3.4.3. Priority Based on Premium and Level of Charge Required

In this case, the priority matrix $Q$ is derived using a combination of both price $P$ and the level of required charge $L$ according to Equation (8).

$$[Q] \sim [P][L]^{-2} \tag{8}$$

## 4. Case Study

Three different cases studies are now investigated to explore the effectiveness of the methodology in the case of EVs sharing a power limited source. A total of 10 EVs are considered in each case. Their battery electrical parameters are summarized in Table 3. These data were used throughout this paper. The circuit diagram used in the simulations is shown in Figure 2.

**Table 3.** Some EV electric parameters [1].

| Model | Battery Voltage (V) | Battery Capacity (kWh) |
|---|---|---|
| Tesla S | 375 | 100 |
| Jaguar I-Pace | 390 | 90 |
| Nissan Leaf | 360 | 62 |
| Citroën C-Zero | 330 | 16 |
| VW e-up | 370 | 18.7 |
| BMW i3 | 353 | 42.2 |
| Mercedes Benz B-class electric | 240 | 28 |
| Ford Focus Electric | 325 | 23 |
| Fiat 500e | 364 | 24 |
| Renault Zoe | 400 | 52 |

[1] https://ev-database.uk/ (accessed on 10 July 2021).

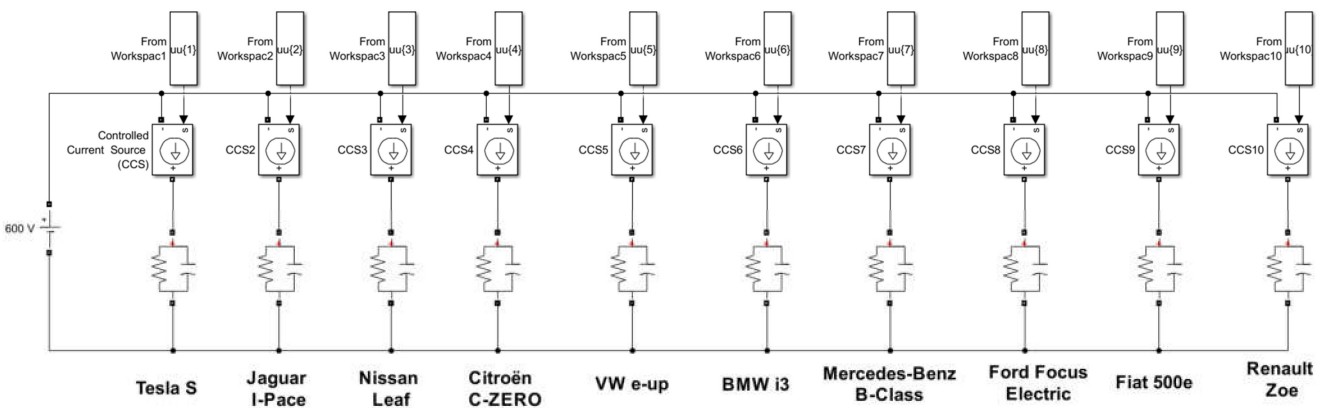

**Figure 2.** Circuit diagram with 10 cars.

For simplicity, it is assumed that the initial terminal voltage of the battery (assumed to be related to State of Charge (SoC)) is 79% of that of a nominal fully charged battery $V_{ref}$ at the start of the scheduled charging trials—although it should be noted that this initial condition is completely arbitrary as far as the proposed algorithm is concerned. In the MPC algorithm, appropriate priority is attributed to each vehicle entering the bay. High priority is given to the vehicles starting to charge, while very low priority is given to the other vehicles on standby. To stop the charging of a vehicle, its priority is switched to a very low value.

*Note*: At all times, the total current allowed for charging is limited to 200 A by the MPC algorithm current constraint (4).

### 4.1. Partial Charging

This scenario considers 10 EVs arriving and departing after partially charging their respective batteries at a charging point in a shopping centre containing five parking bays. The arriving cars were given priority in decreasing order according to their arrival time. For a particular example, the relative priority weighting for each EV is given by the diagonal term of the $Q$ matrix below, according to its ranking in Table 3.

$$Q = \begin{pmatrix} {\color{green}200} & 0 & 0 & 0 & 0 & 0 & 0 & 0 & 0 & 0 \\ 0 & {\color{red}44} & 0 & 0 & 0 & 0 & 0 & 0 & 0 & 0 \\ 0 & 0 & {\color{blue}100} & 0 & 0 & 0 & 0 & 0 & 0 & 0 \\ 0 & 0 & 0 & {\color{magenta}2} & 0 & 0 & 0 & 0 & 0 & 0 \\ 0 & 0 & 0 & 0 & {\color{cyan}5} & 0 & 0 & 0 & 0 & 0 \\ 0 & 0 & 0 & 0 & 0 & {\color{gray}0.02} & 0 & 0 & 0 & 0 \\ 0 & 0 & 0 & 0 & 0 & 0 & 38 & 0 & 0 & 0 \\ 0 & 0 & 0 & 0 & 0 & 0 & 0 & {\color{gray}1} & 0 & 0 \\ 0 & 0 & 0 & 0 & 0 & 0 & 0 & 0 & {\color{orange}0.1} & 0 \\ 0 & 0 & 0 & 0 & 0 & 0 & 0 & 0 & 0 & {\color{magenta}0.01} \end{pmatrix}$$

With this included in the MPC algorithm, the result of the following scenario is shown in Figure 3. In this figure and the subsequent ones, the dashed lines in panel A correspond to the nominal desired terminal for each vehicle. The dashed lines in panel B denote the maximum charging currents for each vehicle model.

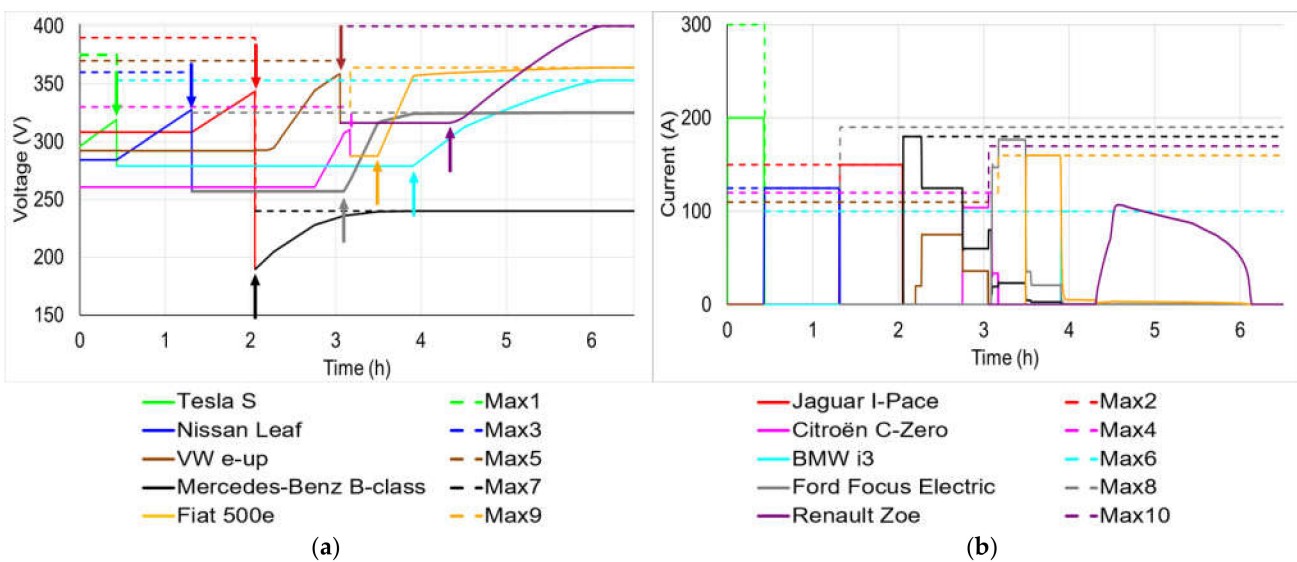

**Figure 3.** Battery terminal voltage (**a**) and current (**b**) of the EVs during partial charging.

Here, the Tesla S (green profile of Figure 3), Nissan Leaf (blue profile), Jaguar I-Pace (red profile), VW e-up (brown profile) and Citroën C-Zero (magenta profile) occupy parking spaces 1, 2, 3, 4 and 5 respectively. At t = 0, the Tesla S begins to charge using the maximum available charging current of 200 A. At 0.43 h, its charge reaches 85% and departs from the main car park (Figure 3, green arrow). This is indicated by the level of its charging current reducing to zero, Figure 3B, and bay 1 becoming temporarily available. Shortly after, A BWM i3 (cyan profile) arrives and parks bay 1 at t = 0.48 h but must wait until t = 3.91 h to commence charging its battery (cyan arrow). At t = 0.43 h, the Nissan Leaf starts its battery charging process. At t = 1.31 h, its charging achieves 91% (blue arrow). It exits the main car park making bay 2 temporarily empty. A Ford Focus Electric (grey profile) arrives and uses the bay at t = 1.34 h, but it must wait until t = 3.09 h (grey arrow) to charge its battery. At t = 1.31 h, the Jaguar I-Pace begins charging. This car departs from the main car park when its charging attains 88% at t = 2.05 h (red arrow). At this stage, bay 3 becomes temporarily free. At t = 2.06 h, a Mercedes-Benz B-class Electric (black profile) enters the car park, utilizes the bay, and begins charging immediately (black arrow). The VW e-up begins to charge its battery at t = 2.21 h. At t = 3.05 h, its charging acquires 97%. It leaves the car park, leaving bay 4 temporarily unoccupied (brown arrow). At t = 3.07 h, a Renault Zoe (purple profile) arrives and takes over this space but must wait until t = 4.33 h to begin charging (purple arrow). Meanwhile, the Citroën C-Zero begins charging its battery at

t = 2.76 h. Later, at t = 2.83 h, its charge reaches 94% of its reference (magenta arrow) and departs—bay 5 becomes temporarily vacant. Shortly after, a Fiat 500e (orange profile) arrives at bay 5 but has to wait until t = 3.49 h to begin charging (orange arrow).

### 4.2. Priority Based on Level of Charging

Initially, this scenario again comprises of the set of 10 EVs wishing to use a public charging station. However, due to the limited power resources available, the operation is scheduled and a priority according to the charge required is attributed to each car. The level of charge required is shown in Table 4.

**Table 4.** Car charge level required.

| Car | Battery Capacity (kWh) | Charge Required (kWh) |
|---|---|---|
| Tesla | 100 | 10 |
| Jaguar I-Pace | 90 | 18 |
| Nissan Leaf | 62 | 24.8 |
| Citroën C-Zero | 16 | 0.32 |
| VW e-up | 18.7 | 0.935 |
| BMW i3 | 94 | 31.65 |
| Mercedes Benz B-Class | 28 | 4.2 |
| Ford Focus Electric | 23 | 2.4 |
| Fiat 500e | 24 | 2.76 |
| Renault Zoe | 52 | 46.8 |

This leads to the matrices $L$ and $Q$ $(= L^{-2})$ below

$$
L = \begin{pmatrix}
10 & 0 & 0 & 0 & 0 & 0 & 0 & 0 & 0 & 0 \\
0 & 18 & 0 & 0 & 0 & 0 & 0 & 0 & 0 & 0 \\
0 & 0 & 24.8 & 0 & 0 & 0 & 0 & 0 & 0 & 0 \\
0 & 0 & 0 & 0.32 & 0 & 0 & 0 & 0 & 0 & 0 \\
0 & 0 & 0 & 0 & 0.935 & 0 & 0 & 0 & 0 & 0 \\
0 & 0 & 0 & 0 & 0 & 31.65 & 0 & 0 & 0 & 0 \\
0 & 0 & 0 & 0 & 0 & 0 & 4.2 & 0 & 0 & 0 \\
0 & 0 & 0 & 0 & 0 & 0 & 0 & 2.76 & 0 & 0 \\
0 & 0 & 0 & 0 & 0 & 0 & 0 & 0 & 2.4 & 0 \\
0 & 0 & 0 & 0 & 0 & 0 & 0 & 0 & 0 & 46.8
\end{pmatrix}
$$

$$
Q = \begin{pmatrix}
0.01 & 0 & 0 & 0 & 0 & 0 & 0 & 0 & 0 & 0 \\
0 & 0.003 & 0 & 0 & 0 & 0 & 0 & 0 & 0 & 0 \\
0 & 0 & 0.002 & 0 & 0 & 0 & 0 & 0 & 0 & 0 \\
0 & 0 & 0 & 9.766 & 0 & 0 & 0 & 0 & 0 & 0 \\
0 & 0 & 0 & 0 & 1.144 & 0 & 0 & 0 & 0 & 0 \\
0 & 0 & 0 & 0 & 0 & 0.001 & 0 & 0 & 0 & 0 \\
0 & 0 & 0 & 0 & 0 & 0 & 0.057 & 0 & 0 & 0 \\
0 & 0 & 0 & 0 & 0 & 0 & 0 & 0.131 & 0 & 0 \\
0 & 0 & 0 & 0 & 0 & 0 & 0 & 0 & 0.174 & 0 \\
0 & 0 & 0 & 0 & 0 & 0 & 0 & 0 & 0 & 0.0005
\end{pmatrix}
$$

The above $Q$ matrix indicates that the Citroën C-Zero features the highest priority coefficient (*PC*) of 9.766. It begins to charge its battery first since it features the least level of charge (*LoC*) with 0.32 kWh, as shown in Table 4. It is followed by the VW e-up (*PC* = 1.144) with the second-lowest level of charging (*LoC* = 0.935) at t = 0.38 h. The Fiat 500e (*PC* = 0.174 with *LoC* = 2.4), Ford Focus Electric (*PC* = 0.1736 with *LoC* = 2.4), Mercedes-Benz B-class Electric (*PC* = 0.057 with *LoC* = 4.2), Tesla S (*PC* = 0.01 and *LoC* = 10), Jaguar I-Pace (*PC* = 0.003 and *LoC* = 18), Nissan Leaf (*PC* = 0.002 and *LoC* = 24.8), BMW i3 (*PC* = 0.001 and *LoC* = 31.65) and Renault Zoe (*PC* = 0.0005 and *LoC* = 46.8) follow

accordingly, with their charging starting at t = 0.83, 1.03, 1.58, 2.32, 3.95, 5.23, 5.71 and 7.89 h, respectively—see Figure 4.

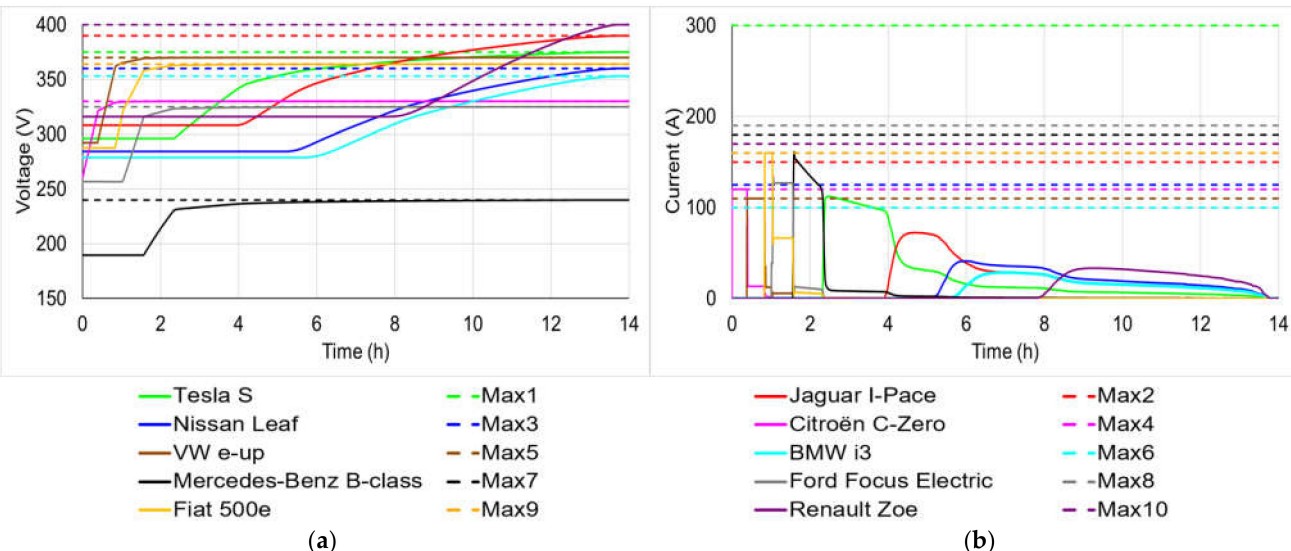

**Figure 4.** Battery terminal voltage (**a**) and current (**b**) of the EVs during charging for the first scenario.

A second scenario consists of the same set of cars as above, except that four of them (Tesla S, Jaguar I-Pace, Citroën C-Zero, Fiat 500e) request the same amount of charge, 16 kWh (see Table 5).

**Table 5.** Four cars of different models require the same amount of charge.

| Car | Battery Capacity (kWh) | Charge Required (kWh) |
| --- | --- | --- |
| Tesla | 100 | 16 |
| Jaguar I-Pace | 90 | 16 |
| Nissan Leaf | 62 | 24.8 |
| Citroën C-Zero | 16 | 16 |
| VW e-up | 18.7 | 0.935 |
| BMW i3 | 94 | 31.65 |
| Mercedes Benz B-Class | 28 | 4.2 |
| Ford Focus Electric | 23 | 2.4 |
| Fiat 500e | 24 | 16 |
| Renault Zoe | 52 | 46.8 |

The matrices $L$ and $Q$ are now given as

$$L = \begin{pmatrix} 16 & 0 & 0 & 0 & 0 & 0 & 0 & 0 & 0 & 0 \\ 0 & 16 & 0 & 0 & 0 & 0 & 0 & 0 & 0 & 0 \\ 0 & 0 & 24.8 & 0 & 0 & 0 & 0 & 0 & 0 & 0 \\ 0 & 0 & 0 & 16 & 0 & 0 & 0 & 0 & 0 & 0 \\ 0 & 0 & 0 & 0 & 0.935 & 0 & 0 & 0 & 0 & 0 \\ 0 & 0 & 0 & 0 & 0 & 31.65 & 0 & 0 & 0 & 0 \\ 0 & 0 & 0 & 0 & 0 & 0 & 4.2 & 0 & 0 & 0 \\ 0 & 0 & 0 & 0 & 0 & 0 & 0 & 2.76 & 0 & 0 \\ 0 & 0 & 0 & 0 & 0 & 0 & 0 & 0 & 16 & 0 \\ 0 & 0 & 0 & 0 & 0 & 0 & 0 & 0 & 0 & 46.8 \end{pmatrix}$$

$$Q = \begin{pmatrix} 0.004 & 0 & 0 & 0 & 0 & 0 & 0 & 0 & 0 & 0 \\ 0 & 0.004 & 0 & 0 & 0 & 0 & 0 & 0 & 0 & 0 \\ 0 & 0 & 0.002 & 0 & 0 & 0 & 0 & 0 & 0 & 0 \\ 0 & 0 & 0 & 0.004 & 0 & 0 & 0 & 0 & 0 & 0 \\ 0 & 0 & 0 & 0 & 1.144 & 0 & 0 & 0 & 0 & 0 \\ 0 & 0 & 0 & 0 & 0 & 0.001 & 0 & 0 & 0 & 0 \\ 0 & 0 & 0 & 0 & 0 & 0 & 0.057 & 0 & 0 & 0 \\ 0 & 0 & 0 & 0 & 0 & 0 & 0 & 0.131 & 0 & 0 \\ 0 & 0 & 0 & 0 & 0 & 0 & 0 & 0 & 0.004 & 0 \\ 0 & 0 & 0 & 0 & 0 & 0 & 0 & 0 & 0 & 0.0005 \end{pmatrix}$$

The above $Q$ matrix shows that the VW e-up with the highest priority coefficient (of 1.144) begins to charge first, since it features the least level of charge of 0.935 kWh, as shown in Table 5. It is followed by the Ford Focus Electric ($PC = 0.131$), with the second lowest level of charging ($LoC = 2.76$) at t = 0.48 h. The Mercedes-Benz B-class Electric ($PC = 0.057$ with $LoC = 4.2$) starts its charging process at t = 0.84 h. The Tesla S, Jaguar I-Pace, Citroën C-Zero and Fiat 500e, with the identical output priority coefficient value of 0.004 and the same charging level of 16 kWh, begin to charge at different points. The Nissan Leaf ($PC = 0.002$ and $LoC = 24.8$), BMW i3 ($PC = 0.001$ and $LoC = 31.65$) and Renault Zoe ($PC = 0.0005$ and $LoC = 46.8$) begin charging at t = 4.44, 5.06 and 7.55 h, respectively—see Figure 5.

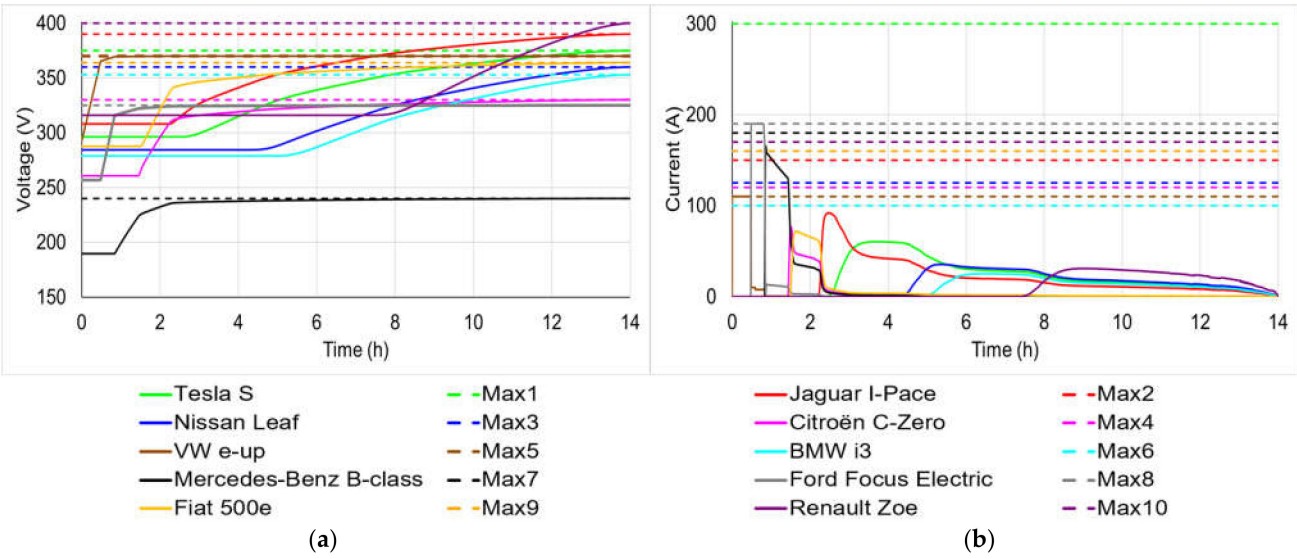

**Figure 5.** Battery terminal voltage (**a**) and current (**b**) of the EVs during charging for this second scenario.

Considering specifically the four cars requesting the same amount of charge, it is relevant to look at their starting sequence (relevant data is shown in Table 6).

**Table 6.** Car data for requests for the same amount of charge.

| Model | Capacitance (F) | Battery Capacity (kWh) | Level of Charge (kWh) |
|---|---|---|---|
| Tesla S | 13,621 | 100 | 16 |
| Jaguar I-Pace | 11,334 | 90 | 16 |
| Citroën C-Zero | 2814 | 16 | 16 |
| Fiat 500e | 3470 | 24 | 16 |

The vehicle with the lowest battery capacity begins to charge first. In this case, Figure 6 shows that the Citroën C-Zero begins to charge at t = 1.44 h (magenta profile). The Fiat

500e (orange profile), Jaguar I-Pace (red profile) and Tesla S (green profile) begin charging their battery at t = 1.5, 2.25 and 2.6 h, respectively.

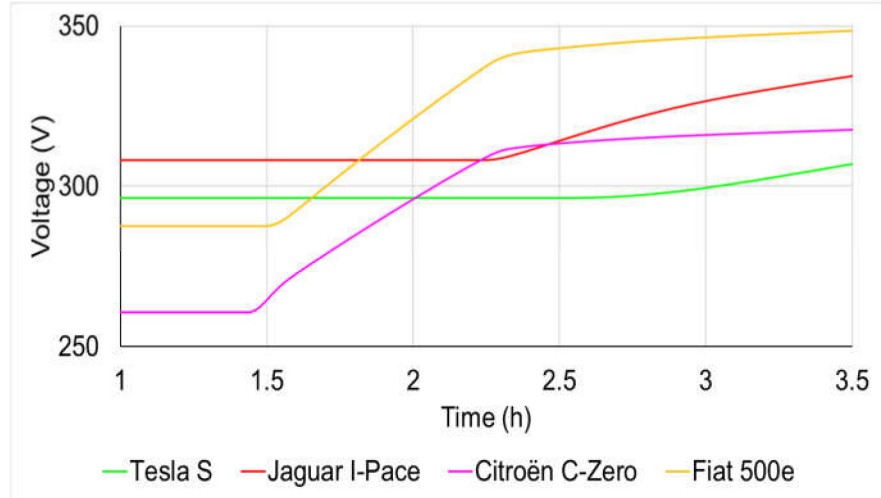

**Figure 6.** Charging profiles of the four cars requesting the same amount of charge.

A third scenario now considers ten cars, except that two of them are the same but request different amounts of charge (a more common scenario). The data for this is shown in Table 7.

**Table 7.** Two identical VW e-up requesting a different amount of charge.

| Car | Battery Capacity (kWh) | Charge Required (kWh) |
|---|---|---|
| Tesla | 100 | 10 |
| Jaguar I-Pace | 90 | 18 |
| Nissan Leaf | 62 | 24.8 |
| Citroën C-Zero | 16 | 16 |
| **VW e-up** | **18.7** | **0.935** |
| BMW i3 | 94 | 31.65 |
| **VW e-up** | **18.7** | **6.92** |
| Ford Focus Electric | 23 | 2.4 |
| Fiat 500e | 24 | 2.76 |
| Renault Zoe | 52 | 46.8 |

The matrices $L$ and $Q$ are therefore chosen as:

$$L = \begin{pmatrix} 10 & 0 & 0 & 0 & 0 & 0 & 0 & 0 & 0 & 0 \\ 0 & 18 & 0 & 0 & 0 & 0 & 0 & 0 & 0 & 0 \\ 0 & 0 & 24.8 & 0 & 0 & 0 & 0 & 0 & 0 & 0 \\ 0 & 0 & 0 & 0.32 & 0 & 0 & 0 & 0 & 0 & 0 \\ 0 & 0 & 0 & 0 & 0.935 & 0 & 0 & 0 & 0 & 0 \\ 0 & 0 & 0 & 0 & 0 & 31.65 & 0 & 0 & 0 & 0 \\ 0 & 0 & 0 & 0 & 0 & 0 & 6.92 & 0 & 0 & 0 \\ 0 & 0 & 0 & 0 & 0 & 0 & 0 & 2.76 & 0 & 0 \\ 0 & 0 & 0 & 0 & 0 & 0 & 0 & 0 & 2.4 & 0 \\ 0 & 0 & 0 & 0 & 0 & 0 & 0 & 0 & 0 & 46.8 \end{pmatrix}$$

$$Q = \begin{pmatrix} 0.01 & 0 & 0 & 0 & 0 & 0 & 0 & 0 & 0 & 0 \\ 0 & 0.003 & 0 & 0 & 0 & 0 & 0 & 0 & 0 & 0 \\ 0 & 0 & 0.002 & 0 & 0 & 0 & 0 & 0 & 0 & 0 \\ 0 & 0 & 0 & 9.766 & 0 & 0 & 0 & 0 & 0 & 0 \\ 0 & 0 & 0 & 0 & 1.144 & 0 & 0 & 0 & 0 & 0 \\ 0 & 0 & 0 & 0 & 0 & 0.001 & 0 & 0 & 0 & 0 \\ 0 & 0 & 0 & 0 & 0 & 0 & 0.021 & 0 & 0 & 0 \\ 0 & 0 & 0 & 0 & 0 & 0 & 0 & 0.131 & 0 & 0 \\ 0 & 0 & 0 & 0 & 0 & 0 & 0 & 0 & 0.174 & 0 \\ 0 & 0 & 0 & 0 & 0 & 0 & 0 & 0 & 0 & 0.0005 \end{pmatrix}$$

The above $Q$ matrix indicates that the Citroën C-Zero features the highest priority coefficient of 9.766. It begins to charge its battery first, since it features the lowest level of charge of 0.32 kWh as shown in Table 7. The first VW e-up ($PC$ = 1.144) with the second lowest level of charge ($LoC$ = 0.935) begins charging its battery at t = 0.38 h. The Fiat 500e ($PC$ = 1.144 with $LoC$ = 2.4) and Ford Focus Electric ($PC$ = 0.131 with $LoC$ = 2.76) start their charging process at t = 0.83 and 1.01 h, respectively. The second VW e-up ($PC$ = 1.144 with $LoC$ = 6.92) starts to charge its battery at t = 1.47 h. The Tesla S ($PC$ = 0.01 with $LoC$ = 10), Jaguar I-Pace ($PC$ = 0.003 with $LoC$ = 18), Nissan Leaf ($PC$ = 0.002 with $LoC$ = 24.8), BMW i3 ($PC$ = 0.001 with $LoC$ = 31.65) and Renault Zoe ($PC$ = 0.0005 and $LoC$ = 46.8) begin their battery charging operation at t = 1.95, 3.36, 4.53, 4.98, and 6.94 h, respectively. All these results are shown in Figure 7.

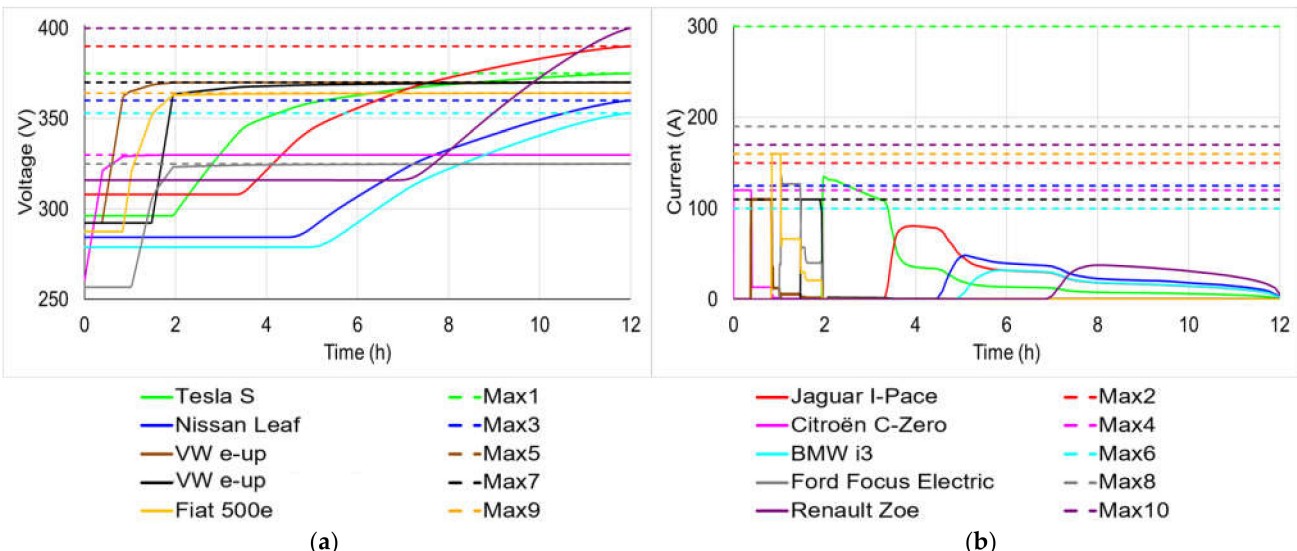

**Figure 7.** Battery terminal voltage (**a**) and current (**b**) of the EVs during charging for the third scenario.

The fourth scenario here involves the same set of cars as in the previous case, except for the two identical VW e-up requiring the same amount of charge. This is shown in Table 8.

**Table 8.** Two identical VW e-up requesting the same amount of charge.

| Car | Battery Capacity (kWh) | Charge Required (kWh) |
| :---: | :---: | :---: |
| Tesla | 100 | 10 |
| Jaguar I-Pace | 90 | 18 |
| Nissan Leaf | 62 | 24.8 |
| Citroën C-Zero | 16 | 0.32 |
| **VW e-up** | **18.7** | **0.935** |
| BMW i3 | 94 | 31.65 |
| **VW e-up** | **18.7** | **0.935** |
| Ford Focus Electric | 23 | 2.4 |
| Fiat 500e | 24 | 2.76 |
| Renault Zoe | 52 | 46.8 |

This corresponds to the matrices $L$ and $Q$ as shown below.

$$
L = \begin{pmatrix}
10 & 0 & 0 & 0 & 0 & 0 & 0 & 0 & 0 & 0 \\
0 & 18 & 0 & 0 & 0 & 0 & 0 & 0 & 0 & 0 \\
0 & 0 & 24.8 & 0 & 0 & 0 & 0 & 0 & 0 & 0 \\
0 & 0 & 0 & 0.32 & 0 & 0 & 0 & 0 & 0 & 0 \\
0 & 0 & 0 & 0 & 0.935 & 0 & 0 & 0 & 0 & 0 \\
0 & 0 & 0 & 0 & 0 & 31.65 & 0 & 0 & 0 & 0 \\
0 & 0 & 0 & 0 & 0 & 0 & 0.935 & 0 & 0 & 0 \\
0 & 0 & 0 & 0 & 0 & 0 & 0 & 2.76 & 0 & 0 \\
0 & 0 & 0 & 0 & 0 & 0 & 0 & 0 & 2.4 & 0 \\
0 & 0 & 0 & 0 & 0 & 0 & 0 & 0 & 0 & 46.8
\end{pmatrix}
$$

$$
Q = \begin{pmatrix}
0.01 & 0 & 0 & 0 & 0 & 0 & 0 & 0 & 0 & 0 \\
0 & 0.003 & 0 & 0 & 0 & 0 & 0 & 0 & 0 & 0 \\
0 & 0 & 0.002 & 0 & 0 & 0 & 0 & 0 & 0 & 0 \\
0 & 0 & 0 & 9.766 & 0 & 0 & 0 & 0 & 0 & 0 \\
0 & 0 & 0 & 0 & 1.144 & 0 & 0 & 0 & 0 & 0 \\
0 & 0 & 0 & 0 & 0 & 0.001 & 0 & 0 & 0 & 0 \\
0 & 0 & 0 & 0 & 0 & 0 & 1.144 & 0 & 0 & 0 \\
0 & 0 & 0 & 0 & 0 & 0 & 0 & 0.131 & 0 & 0 \\
0 & 0 & 0 & 0 & 0 & 0 & 0 & 0 & 0.174 & 0 \\
0 & 0 & 0 & 0 & 0 & 0 & 0 & 0 & 0 & 0.0005
\end{pmatrix}
$$

The above $Q$ matrix indicates that the Citroën C-Zero (highest priority coefficient of 9.766) begins to charge its battery first, since it has required the least level of charge of 0.32 kWh as shown in Table 8. The two VW e-ups, where each has $PC$ = 1.144 with 0.935 kWh, both commence charging their battery at t = 0.38 h simultaneously (black profile). The Fiat 500e ($PC$ = 0.174 with $LoC$ = 2.4), Ford Focus Electric ($PC$ = 0.131 with $LoC$ = 2.76), Tesla S ($PC$ = 0.01 with $LoC$ = 10), Jaguar I-Pace ($PC$ = 0.003 with $LoC$ = 18), Nissan Leaf ($PC$ = 0.002 with $LoC$ = 24.8), BMW i3 ($PC$ = 0.001 with $LoC$ = 31.65) and Renault Zoe ($PC$ = 0.0005 and $LoC$ = 46.8) begin their battery charging process at t = 0.92, 1.1, 1.75, 3.15, 4.32, 4.81 and 6.73 h, respectively—see Figure 8.

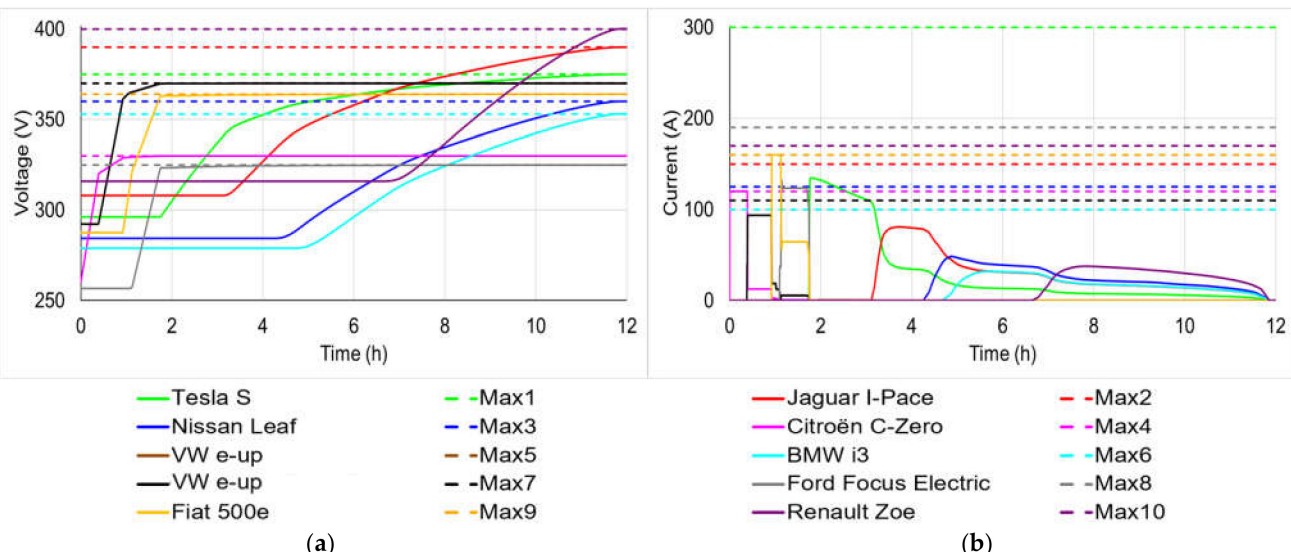

**Figure 8.** Battery terminal voltage (**a**) and current (**b**) of the EVs during charging for the fourth scenario.

### 4.3. Priority Based on Price Premium

This scenario consists of the same set above at a public station where the standard price is 30 p/kWh. Due to the limited power resource, charging is scheduled and a priority according to the price the customer is willing to pay is attributed to each car. The prices customers are willing to pay are (arbitrarily) summarized in Table 9.

**Table 9.** Price customer is inclined to pay.

| Customer | Price (p) |
|---|---|
| Tesla | 45 |
| Jaguar I-Pace | 30 |
| Nissan Leaf | 68 |
| Citroën C-Zero | 94 |
| VW e-up | 100 |
| BMW i3 | 48 |
| Mercedes Benz B-Class | 42 |
| Ford Focus Electric | 75 |
| Fiat 500e | 80 |
| Renault Zoe | 72 |

This corresponds to the matrices $P$ and $Q$ (= $P$) below.

$$
Q = P = \begin{pmatrix}
45 & 0 & 0 & 0 & 0 & 0 & 0 & 0 & 0 & 0 \\
0 & 30 & 0 & 0 & 0 & 0 & 0 & 0 & 0 & 0 \\
0 & 0 & 68 & 0 & 0 & 0 & 0 & 0 & 0 & 0 \\
0 & 0 & 0 & 94 & 0 & 0 & 0 & 0 & 0 & 0 \\
0 & 0 & 0 & 0 & 100 & 0 & 0 & 0 & 0 & 0 \\
0 & 0 & 0 & 0 & 0 & 48 & 0 & 0 & 0 & 0 \\
0 & 0 & 0 & 0 & 0 & 0 & 42 & 0 & 0 & 0 \\
0 & 0 & 0 & 0 & 0 & 0 & 0 & 75 & 0 & 0 \\
0 & 0 & 0 & 0 & 0 & 0 & 0 & 0 & 80 & 0 \\
0 & 0 & 0 & 0 & 0 & 0 & 0 & 0 & 0 & 72
\end{pmatrix}
$$

The above $Q$ matrix shows that the VW e-up (highest priority coefficient of 100) begins to charge its battery first, since it has paid the highest price ($P$), of 100, as shown in Table 9. The Citroën C-Zero with the second highest priority coefficient of 94 and $P$ = 94

begins charging its battery at t = 0.07 h. The Fiat 500e (*PC* = *P* = 80), Ford Focus Electric (*PC* = *P* = 75), Renault Zoe (*PC* = *P* = 72), Nissan Leaf (*PC* = *P* = 68), BMW i3 (*PC* = *P* = 48), Tesla S (*PC* = *P* = 45), Mercedes-Benz B-class (*PC* = *P* = 42) and Jaguar I-Pace (*PC* = *P* = 30) start their battery charging process at t = 0.21, 0.44, 0.63, 1.24, 1.29, 2.56, 2.89 and 3.04 h, respectively. The result for this case is shown in Figure 9.

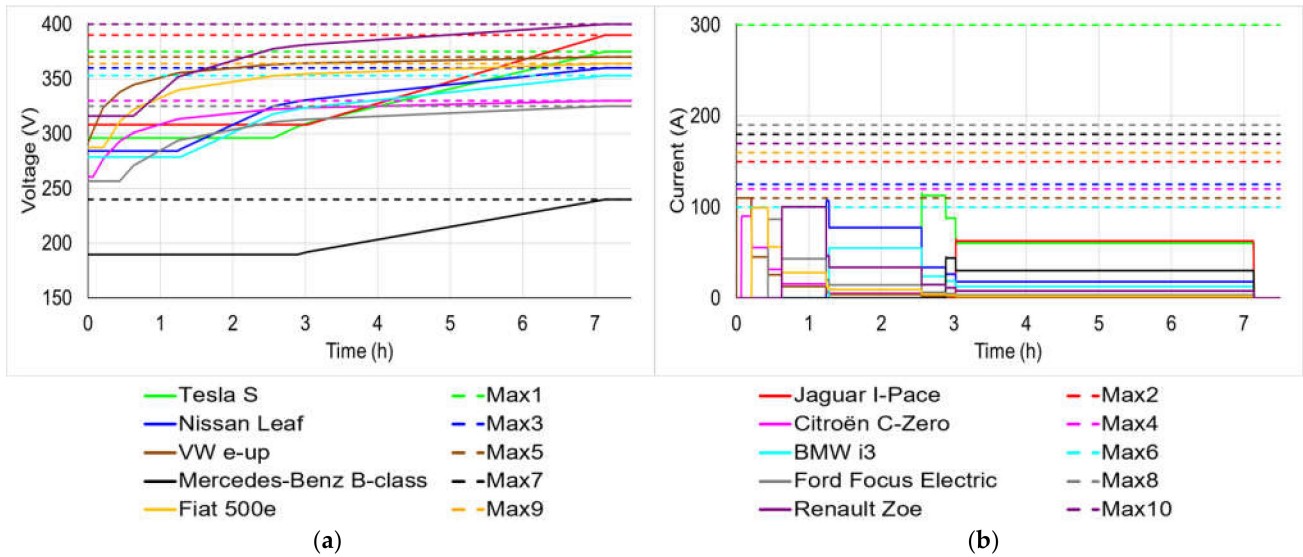

**Figure 9.** Battery terminal voltage (**a**) and current (**b**) of the EVs during charging using price premium priority.

### 4.4. Priority Based on Premium and Level of the Charge Request

This scenario considers the same set of cars as previously and applies the last-priority allocation $[P][L]^{-2}$ with two of the EVs having different premium prices $p_1$ and $p_2$ and levels of charge request $L_1$ and $L_2$ such that $p_1/L_1^2 \approx p_2/L_2^2$. The level of charge required and the price used in the simulation are shown in Table 10.

**Table 10.** Car charge level required.

| Car | Battery Capacity (kWh) | Charge Required (kWh) | Price (p) |
|---|---|---|---|
| Tesla | 100 | 10 | 72 |
| Jaguar I-Pace | 90 | 18 | 75 |
| Nissan Leaf | 62 | 24.8 | 80 |
| Citroën C-Zero | 16 | 0.32 | 30 |
| VW e-up | 18.7 | 0.935 | 42 |
| BMW i3 | 94 | 31.65 | 130 |
| Mercedes Benz B-Class | 28 | 4.2 | 68 |
| Ford Focus Electric | 23 | 2.4 | 48 |
| Fiat 500e | 24 | 2.76 | 45 |
| Renault Zoe | 52 | 46.8 | 100 |

These lead to the matrices $P$, $L$ and $Q$ ($= PL^{-2}$) below, where the diagonal entries of the Nissan Leaf and BMW i3 in the $Q$ matrix are very close.

$$
P = \begin{pmatrix}
72 & 0 & 0 & 0 & 0 & 0 & 0 & 0 & 0 & 0 \\
0 & 75 & 0 & 0 & 0 & 0 & 0 & 0 & 0 & 0 \\
0 & 0 & 80 & 0 & 0 & 0 & 0 & 0 & 0 & 0 \\
0 & 0 & 0 & 30 & 0 & 0 & 0 & 0 & 0 & 0 \\
0 & 0 & 0 & 0 & 42 & 0 & 0 & 0 & 0 & 0 \\
0 & 0 & 0 & 0 & 0 & 130 & 0 & 0 & 0 & 0 \\
0 & 0 & 0 & 0 & 0 & 0 & 68 & 0 & 0 & 0 \\
0 & 0 & 0 & 0 & 0 & 0 & 0 & 48 & 0 & 0 \\
0 & 0 & 0 & 0 & 0 & 0 & 0 & 0 & 45 & 0 \\
0 & 0 & 0 & 0 & 0 & 0 & 0 & 0 & 0 & 100
\end{pmatrix}
$$

$$
L = \begin{pmatrix}
10 & 0 & 0 & 0 & 0 & 0 & 0 & 0 & 0 & 0 \\
0 & 18 & 0 & 0 & 0 & 0 & 0 & 0 & 0 & 0 \\
0 & 0 & 24.8 & 0 & 0 & 0 & 0 & 0 & 0 & 0 \\
0 & 0 & 0 & 0.32 & 0 & 0 & 0 & 0 & 0 & 0 \\
0 & 0 & 0 & 0 & 0.935 & 0 & 0 & 0 & 0 & 0 \\
0 & 0 & 0 & 0 & 0 & 31.65 & 0 & 0 & 0 & 0 \\
0 & 0 & 0 & 0 & 0 & 0 & 4.2 & 0 & 0 & 0 \\
0 & 0 & 0 & 0 & 0 & 0 & 0 & 2.76 & 0 & 0 \\
0 & 0 & 0 & 0 & 0 & 0 & 0 & 0 & 2.4 & 0 \\
0 & 0 & 0 & 0 & 0 & 0 & 0 & 0 & 0 & 46.8
\end{pmatrix}
$$

$$
Q = \begin{pmatrix}
0.72 & 0 & 0 & 0 & 0 & 0 & 0 & 0 & 0 & 0 \\
0 & 0.232 & 0 & 0 & 0 & 0 & 0 & 0 & 0 & 0 \\
0 & 0 & 0.130 & 0 & 0 & 0 & 0 & 0 & 0 & 0 \\
0 & 0 & 0 & 292.97 & 0 & 0 & 0 & 0 & 0 & 0 \\
0 & 0 & 0 & 0 & 48.04 & 0 & 0 & 0 & 0 & 0 \\
0 & 0 & 0 & 0 & 0 & 0.1298 & 0 & 0 & 0 & 0 \\
0 & 0 & 0 & 0 & 0 & 0 & 3.855 & 0 & 0 & 0 \\
0 & 0 & 0 & 0 & 0 & 0 & 0 & 6.30 & 0 & 0 \\
0 & 0 & 0 & 0 & 0 & 0 & 0 & 0 & 7.813 & 0 \\
0 & 0 & 0 & 0 & 0 & 0 & 0 & 0 & 0 & 0.046
\end{pmatrix}
$$

The above $Q$ matrix indicates that the Citroën C-Zero ($Q = PC = 292.97$) starts charging its battery first. It is followed by VW e-up ($Q = PC = 48.04$), Fiat 500e ($Q = PC = 7.813$), Ford Focus Electric ($Q = PC = 6.30$), Mercedes Benz B-Class ($Q = PC = 3.855$), Tesla S ($Q = PC = 0.72$) and Jaguar I-Pace ($Q = PC = 0.232$). For the two cars, BMW i3 and Nissan Leaf, with the close diagonal entry values ($Q = PC \approx 0.130$), the former with the smallest battery capacitance (6487 F), starts charging before the latter (9163 F). The BMW i3 starts at 3.3 h, while the Nissan Leaf begins at 3.9 h. The Renault Zoe, with the smallest diagonal entry ($Q = PC = 0.046$) in the matrix $Q$, starts its charging operation at 5.13 h. These are shown in Figure 10, with a close view in panel Figure 10b.

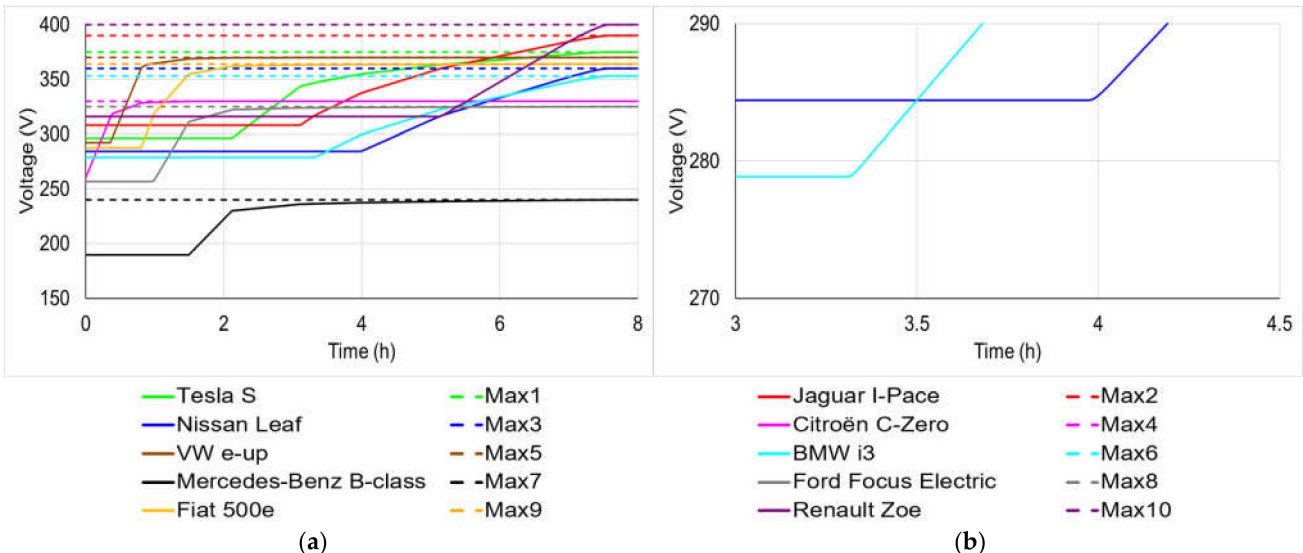

**Figure 10.** Battery terminal voltage (**a**,**b**) a close view of the BMW i3 and Nissan Leaf voltage.

## 5. Impact of Additional Local Grid Load (e.g., Domestic Housing Estate)

Switching in and out of additional loads will generate voltage disturbances at the junction between the charging bays and the additional local loads (e.g., domestic supply). The level of disturbance depends on many factors, such as the strength of the grid or the size of the switched load. This section aims to investigate the impact of load switching on the EV charger installation, and how it impacts the charging scheduling in this case. All previous scenarios considered were performed assuming the availability of a stiff power source. In this section, the EV charging bays are connected to a grid system through a 33/0.444 kV–275 kVA transformer. Three cars are charged as an example: The Tesla S, Jaguar I-Pace and Nissan Leaf. In this scenario, the impact of load switching at a nodal point of the EV charging bays is investigated. The model used for the investigation is shown in Figure 11 and includes the EV charging bays, a small estate load and a filter.

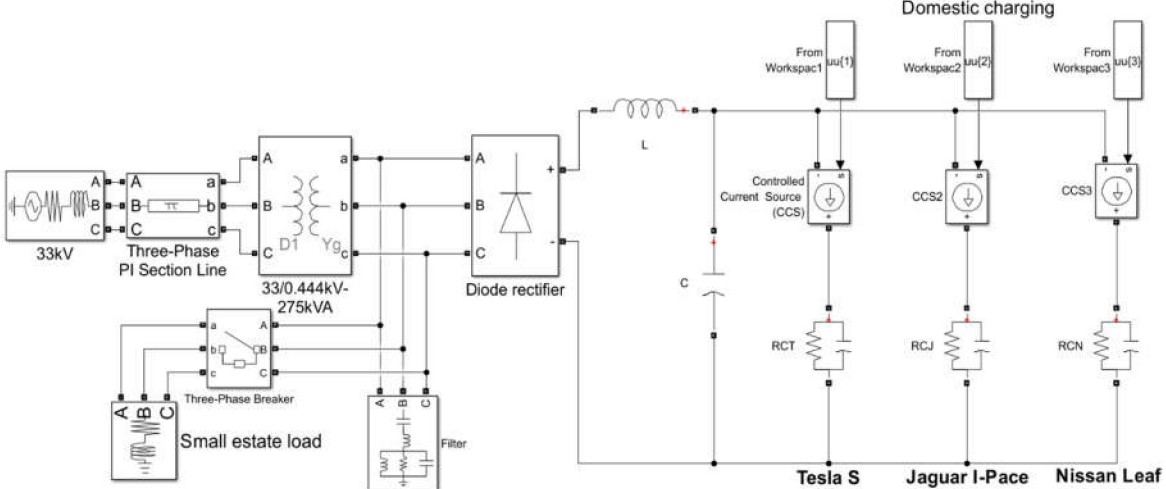

**Figure 11.** Grid system with small estate load.

First, a load with the same apparent power (120 kVA) as the EV charging system is considered. The power factor of the load is taken as $\cos \phi = 0.9$. The load is switched on at 25 s and switched off at 35 s. At these times, the magnitude and frequency of the AC voltage at the nodal point are disturbed—see Figure 12. The magnitude of the voltage dip

depends on the strength of the grid. The level of voltage change, which is less than 6% in this case, is acceptable according to [24]. Furthermore, the frequency, which is perturbed for a short amount of time, remains within the permissible range 49.5 Hz–50.5 Hz, as specified in [27] in this scenario.

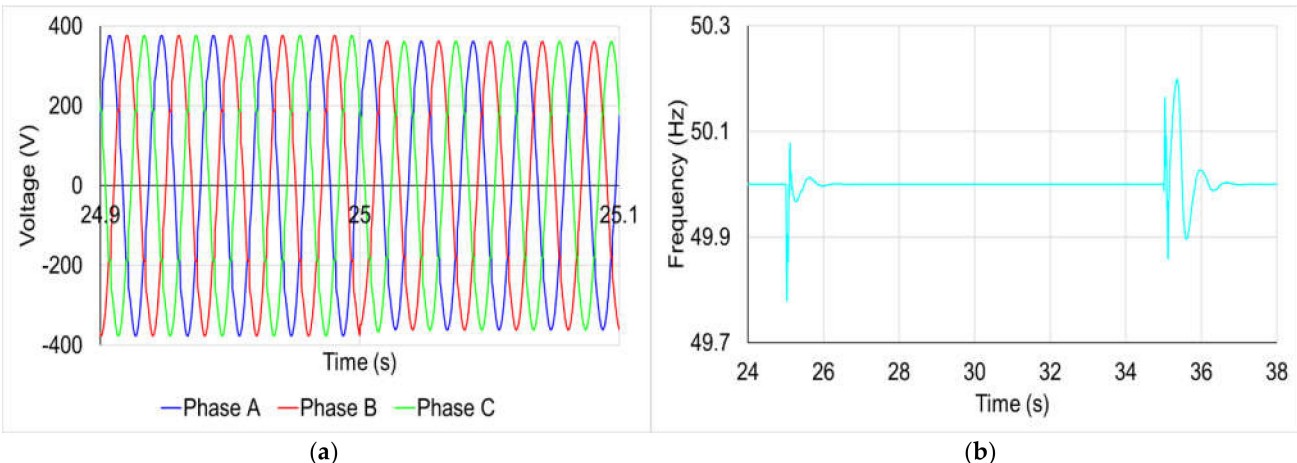

<div align="center">(<strong>a</strong>)               (<strong>b</strong>)</div>

**Figure 12.** (**a**) AC Voltage. (**b**) Frequency fluctuations.

Secondly, the impact of the level of the external load value on the EV charging is investigated. Three values of load are considered. These are 184.4 kW, 5 × 184.4 kW and 6 × 184.4 kW, with 184.4 kW being the maximum available power of IEC 61851 mode 3 [28]. These loads correspond, respectively, to the fast charging of one, five and six cars. The size of the transformer was changed to 1.4 MVA to allow for the largest load 6 × 184.4 kW assuming a power factor of 0.9. The circuit diagram used for this study is presented in Figure 13. For these three loads, the largest magnitude of voltage drop at the point of connection is less than 2%, as shown in Figure 14. This AC voltage drop is reflected on the DC side, depending on the size of the load, (see Table 11 and Figure 15). The graph of the DC current is also presented in Figure 15, which shows that the MPC algorithm maintains the current limit (200 A) despite the voltage fluctuation.

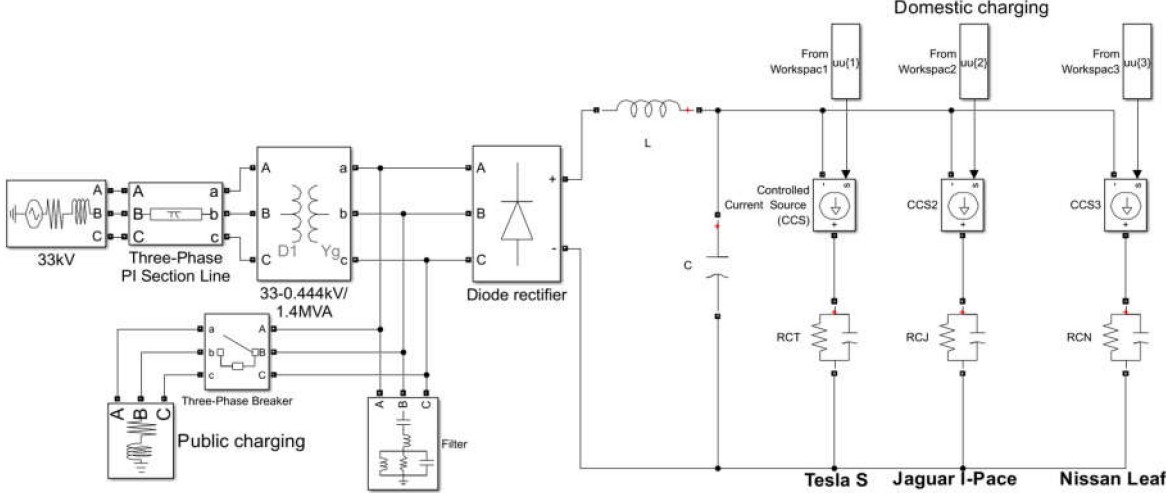

**Figure 13.** Grid system with public charging.

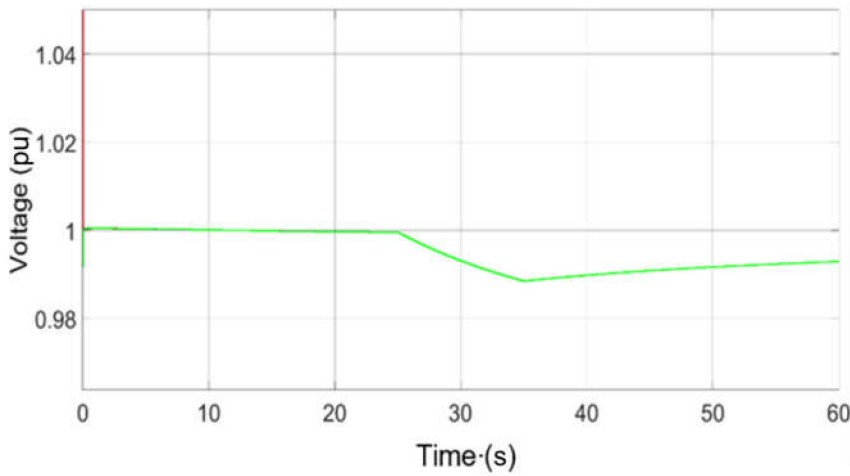

**Figure 14.** Voltage fluctuation at the point of connection.

**Table 11.** Voltage fluctuation at the DC side.

| Charging power (kW) | 184.4 | 5 × 184.4 | 6 × 184.4 |
|---|---|---|---|
| Volt drop (%) | 0.97 | 5.10 | 6.18 |

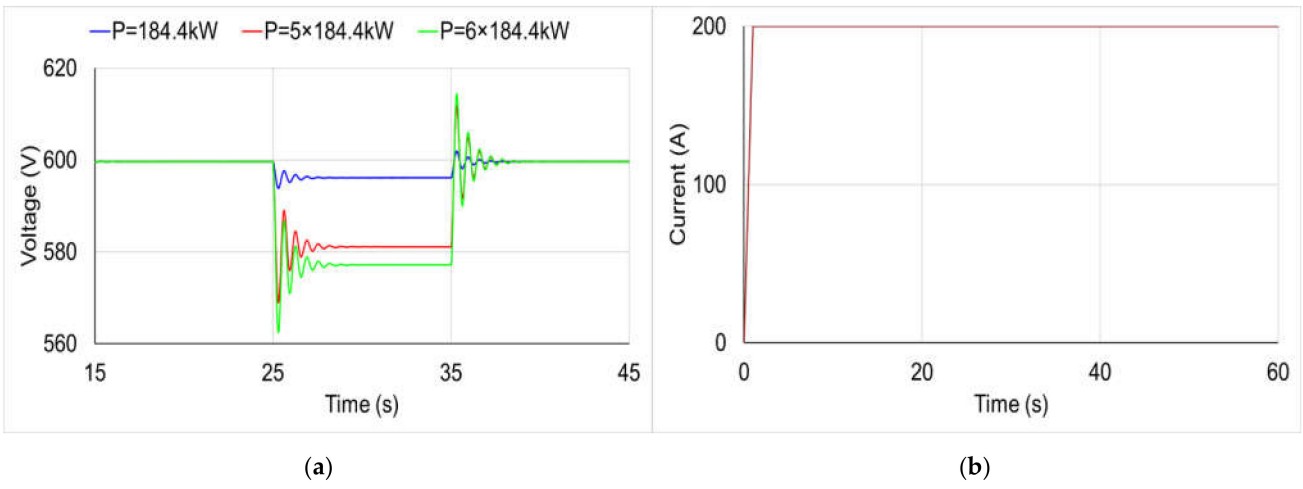

(**a**)                                                                        (**b**)

**Figure 15.** (**a**) DC Voltage (**b**) Current profile.

## 6. Application to Demand-Side Response

The increase in the number of EV charging points to support the expansion of EVs could give rise to one of the largest forms of electricity consumption in the UK. This implies that the total electricity demand fluctuates across the day. Furthermore, if the production of energy relies on renewables, such as wind and solar, which are well known for their intermittency, the balance of available power may not be guaranteed, especially if there are no other sources, such as storage, to cover the demand. Balancing energy is used by system operators to address unplanned fluctuations in the production of electricity or load consumption. The first option is to switch on generators from the production side. In the context of V2G technology, EVs could contribute to this grid balancing by feeding back the energy stored in their battery to the grid. This operation requires the EVs to remain connected to the grid when they are not in circulation. Furthermore, it requires the monitoring of parameters, such as capacity and the state of charge of the battery side. The second option, from the consumption side, is to reduce or switch off loads. As EVs are loads as well during their charging operation, this latter option is chosen to avoid building

power plants that will be used only for a few hours per year. In this part, the amount of current charging in a fleet of EVs is temporarily reduced from 200 to 5 A to balance the energy used (see Figure 16).

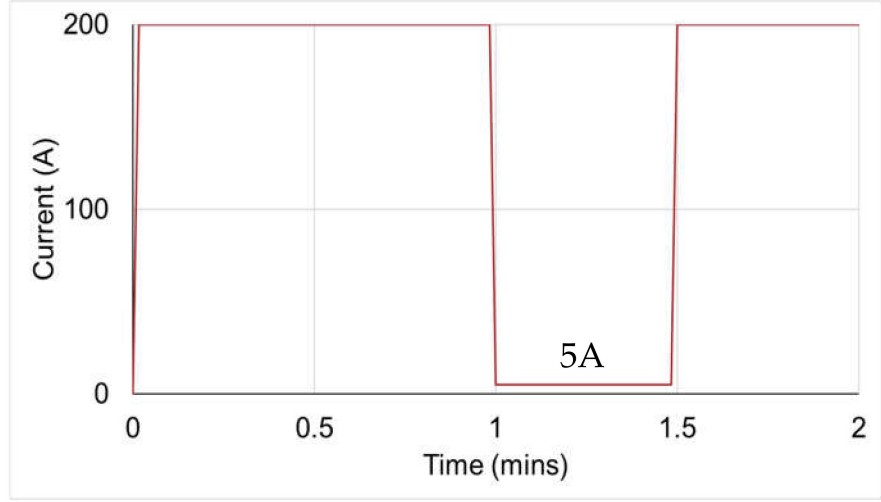

**Figure 16.** Demand-side response applied to EV charging.

## 7. Discussion

Control of the scheduling of EV charging in case of limited power resources was presented, based on MPC. To allow for scheduling, priority was attributed to each EV through the output matrix $Q$ of the quadratic cost function. Three criteria for priority were used. The first is based on the level of charging, the second is based on price premium and the third uses a combination of both. These are examples of criteria that could be used to prevent excessive queuing.

For the criterion based on the level of charge request, the priority is allocated to the EV requiring the lowest charging amount.

- In the case where a set of cars request the same amount of charge, the priority is attributed according to the size of the battery (from lowest to highest).
- In the case where a set of waiting cars contains two identical EVs:
  - the priority between the two is assigned to the car requiring the least amount of charge.
  - when the two request the same amount of charge, both charge their batteries simultaneously.

For the criterion based on price premium, the priority is allocated in descending order of the price that the customer is willing to pay.

The third priority attribution features the two components: price premium and level of charge requested. The charging operation occurs in descending order of the priority specified in the diagonal entry of the quadratic form matrix. When two or more EVs feature very close or equal diagonal entry values, the charging order is decided according to the size of the battery capacitance, in ascending order.

As the ranges of the price and the level of charge request are not anticipated to be too large (at a public charging station, the price difference would not be enormous and the range of EV battery capacitance would not be excessive), the charging priority attributed by this formula is according to the magnitude of the component $p/L^2$, except for EVs with similar or equal diagonal entry values. This was confirmed by preliminary simulations carried out to test the validity of the "criterion". Furthermore, as the criterion is written, there is no prominence of one criterion over the other. It is suggested that, for cases where two or more variables are involved in the criterion, a prominence ranking be introduced as a weighting factor.

In all cases, it was shown that MPC, with appropriate constraints, can provide a means of priority charge scheduling based on premiums to be paid or required.

The proposed control strategy allows the avoidance of grid overloads, particularly during peak demand periods. Furthermore, in anticipation of the increase in the number of EV charging stations, the method allows the participation of these installations in Demand-Side Response schemes.

## 8. Conclusions

This paper proposed a method based on MPC combined with a multi-agent system to schedule EV charging operation at public stations with a constraint on the available peak energy. The aim was to mitigate some issues that could impact the distribution system in the case of large-scale use of EVs. This includes the avoidance of grid overloads, particularly during peak demand periods.

The MPC is an algorithm that makes it possible to solve multi-variable problems subjected to an input or output constraint. Each EV is represented as an autonomous agent, which interacts with other agents and tries to achieve its own goals. The associated cost function, written as a quadratic form, includes two matrices, $R$ and $Q$. The first, termed the weight input matrix, is used in the control of the EV charging current, while the second, termed the output priority matrix, contains the EV assigned priority. Three priority assignment criteria were tested in the simulation: first, the level of charge required; second, the use of premium price; and third, the combination of both. The simulation results show that the schedule of the charging operation follows the assigned priority, except for cases in which two or more cars feature very close or equal diagonal entry values. In this case, the charging order for these cars is determined according to the ascending size of their battery capacitance.

The proposed control strategy could help to reduce queue lengths and waiting times during the charging process at public stations.

**Author Contributions:** Writing—original draft preparation, M.L.R.; writing—review and supervision, C.B., N.S. and M.B. All authors have read and agreed to the published version of the manuscript.

**Funding:** The Smart Energy Network Demonstrator (SEND) project (ref. 32R16P00706) is part-funded through the European Regional Development Fund (ERDF) as part of the England 2014 to 2020 European Structural and Investment Funds (ESIF) Growth Program, and Power Technologies Limited (PTL, company registration number 08873798). The project also receives funds from the UK Department for Business, Energy and Industrial Strategy (BEIS).

**Institutional Review Board Statement:** Not applicable.

**Informed Consent Statement:** Not applicable.

**Data Availability Statement:** Not applicable.

**Acknowledgments:** The research was supported by SEND, PTL and BEIS.

**Conflicts of Interest:** The funders had no role in the design of the study; in the collection, analyses, or interpretation of data; in the writing of the manuscript, or in the decision to publish the results.

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
