# Peer review of "EV Charging in Case of Limited Power Resource"

_actuators, doi:10.3390/act10120325_

Round 1

Reviewer 1 Report

The paper needs substantial changes to be accepted in this peer-reviewed journal

1-More insight on EV charging strategies is mandatory, kindly refer

Karmaker, A.K., Hossain, M., Manoj Kumar, N., Jagadeesan, V., Jayakumar, A. and Ray, B., 2020. Analysis of Using Biogas Resources for Electric Vehicle Charging in Bangladesh: A Techno-Economic-Environmental Perspective. Sustainability12(7), p.2579.

2-No need of fundamentals like Fig-1

3-What does the author try to say in the section "3.3.3. Priority based on premium and level of charge required"

is not clear

4-The number of references especially recent one are very low for this manuscript.

5-Make a comprehensive proof reading and enhance the clarity of the figure

Reviewer 2 Report

The work tries to propose methods of attribution of charging priority based on level of charge required and premiums. The proposed solution is based on model predictive control (MPC) which maintains total current/power within limits (which can change with time) and which imparts real-time priority charge scheduling of multiple charging bays. The method is also shown to readily allow in participation in Demand Side Response (DSR) schemes. 

The paper is basically well written and offers interesting results. However, there are some issues which must be improved significantly. 

  1. The novelty of the paper needs to be justified and clearly defined. It includes the clear difference with the available literature and previous works, whether on the same case of Denmark and/or other countries.
  2. In the abstract and highlights, please also include the quantitative results, not only qualitative explanation. Therefore, the readers can understand accurately the contents of each used references. 
  3. Many references are lumped together without providing sufficient description for each of them. It gives nothing to the readers. Please provide a short description or descriptor for each used references, hence, the readers can understand the content of each used references. 
  4. The authors must be able to show the correlation with the previous correlated studies and their limitations; therefore, the urgency of current study could be understood and justified well. 
  5. Use SI unit and don't mix the unit standards (use km instead of mile). 
  6. Further deeper explanation about the demand response must be further elaborated. 
  7. In order to improve the correlation of the study and continuity with the previous studies, additional literature study and references are required. Below are several works which are strongly recommended. 
    1. Battery-assisted charging system for simultaneous charging of electric vehicles. Energy 100 (2016) 82-90
    2. Quantifying flexibility of residential electric vehicle charging loads using non-intrusive load extracting algorithm in demand response. Sustainable Cities and Society 50 (2019) 101664
    3. Modeling of plug-in electric vehicle travel patterns and charging load based on trip chain generation. Journal of Power Sources 359 (2017) 468-479. 

Reviewer 3 Report

This paper focuses on public charging stations and proposes methods of attribution of charging priority based on level of charge required and premiums in case of limited power resource. There are several issues that need to be revised.
1.In 1 section, what research has been done in the existing literature in case of limited power resource? Please clarify in detail.
2.In 1 section, please explain the difference between this paper and the previous research. Please explain the innovation of this paper.
3.The combination of both price P and the level of required charge L has different forms, how is the definition of formula (8) obtained? How to prove its effectiveness?
4.The proposed charging strategy is to charge the electric vehicle simultaneously for all available bay, and change the charging current according to the importance and priority.

Round 2

Reviewer 1 Report

Good improvements have been made, kidly consider following

1-Enhance the clarity of the figure and make a comprehensive proofreading

rearrange the sentence

"Widespread use of electric vehicles (EVs) presents benefits to the environment but can also bring, according to Hardman et al [1], significant operational challenges to existing power networks"

2- Kindly add a crisp "Conclusion" part

Reviewer 2 Report

The authors have sufficiently addressed the reviews, as well as improved and revised the manuscript. 

Author Response

Thanks a lot for your help and suggestions.

Reviewer 3 Report

This paper focuses on public charging stations and proposes methods of attribution of charging priority based on level of charge required and premiums in case of limited power resource. There are several issues that need to be revised.
1.In 1 section, what research has been done in the existing literature in case of limited power resource? Please clarify in detail.
2.In 1 section, please explain the difference between this paper and the previous research. Please explain the innovation of this paper.
3.The combination of both price P and the level of required charge L has different forms, how is the definition of formula (8) obtained? How to prove its effectiveness?
4.The proposed charging strategy is to charge the electric vehicle simultaneously for all available bay, and change the charging current according to the importance and priority.

Author Response

All of these points have been already addressed in the previous revision of the  manuscript dated 16 November 2021.

Round 3

Reviewer 3 Report

The author has modified the manuscript according to the comments, but there is another question to be revised:
In 3.4.3, the dimensions of P and L-2 are different and should be normalized to avoid that one variable is too large to weaken the role of other variables.
